# Pyrolytic Behavior of Polyvinyl Chloride: Kinetics, Mechanisms, Thermodynamics, and Artificial Neural Network Application

**DOI:** 10.3390/polym13244359

**Published:** 2021-12-13

**Authors:** Mohammed Al-Yaari, Ibrahim Dubdub

**Affiliations:** Chemical Engineering Department, King Faisal University, P.O. Box 380, Al-Ahsa 31982, Saudi Arabia; malyaari@kfu.edu.sa

**Keywords:** polyvinyl chloride (PVC), pyrolysis, thermogravimetric analysis (TGA), kinetics, thermodynamics, artificial neural networks (ANN)

## Abstract

Pyrolysis of waste polyvinyl chloride (PVC) is considered a promising and highly efficient treatment method. This work aims to investigate the kinetics, and thermodynamics of the process of PVC pyrolysis. Thermogravimetry of PVC pyrolysis at three heating rates (5, 10, and 20 K/min) showed two reaction stages covering the temperature ranges of 490–675 K, and 675–825 K, respectively. Three integral isoconversional models, namely Flynn-Wall-Qzawa (FWO), Kissinger-Akahira-Sunose (KAS), and Starink, were used to obtain the activation energy (*E_a_*), and pre-exponential factor (*A*) of the PVC pyrolysis. On the other hand, the Coats-Redfern non-isoconversional model was used to determine the most appropriate solid-state reaction mechanism/s for both stages. Values of *E_a_*, and *A*, obtained by the isoconversional models, were very close and the average values were, for stage I: *E_a_* = 75 kJ/mol, *A* = 1.81 × 10^6^ min^−1^; for stage II: *E_a_* = 140 kJ/mol, *A* = 4.84 × 10^9^ min^−1^. In addition, while the recommended mechanism of the first stage reaction was P2, F3 was the most suitable mechanism for the reaction of stage II. The appropriateness of the mechanisms was confirmed by the compensation effect. Thermodynamic study of the process of PVC pyrolysis confirmed that both reactions are endothermic and nonspontaneous with promising production of bioenergy. Furthermore, a highly efficient artificial neural network (ANN) model has been developed to predict the weight left % during the PVC pyrolysis as a function of the temperature and heating rate. The 2-10-10-1 topology with TANSIG-LOGSIG transfer function and feed-forward back-propagation characteristics was used.

## 1. Introduction

Plastics are widely used because of their distinguished properties including degradation resistance, flexibility, and low weight and cost [1]. Therefore, the global production rate of plastics is increasing dramatically, and thus massive plastic waste is generated. Unfortunately, most of the plastic waste is either incinerated or disposed of in landfills [2] which causes major environmental concerns. Pyrolysis has been reported as a very promising thermochemical method to treat plastic waste and produce bioenergy and/or valuable chemicals [3,4].

Municipal plastic waste (MPW) mainly comprises low-density polyethylene (LDPE), high-density polyethylene (HDPE), polypropylene (PP), polystyrene (PS), polyethylene terephthalate (PET), polyvinyl chloride (PVC), and other plastics. PVC represents around 11 wt% of the MPW, however, the composition may change from one location to another [5]. In this work, the pyrolysis process of PVC is targeted for investigation.

Kim (2001) [6] studied the pyrolysis of polyvinyl chloride (PVC) using thermogravimetric analysis (TGA) data at three different heating rates (5, 10, and 30 K/min). Two pyrolysis stages were observed and attributed to the production of volatiles and intermediates, respectively. The Freeman-Carroll model was used to obtain the kinetics parameters. The average values of the triple kinetic parameters (activation energy (*E_a_*), pre-exponential factor (*A*), and reaction order (*n*)) were reported as *E_a_* = 129.95 kJ/mol, *A* = 3.04 × 10^11^ min^−1^, and *n* = 1.49 for the first stage, and *E_a_* = 282.05 kJ/mol, *A* = 2.63 × 10^20^ min, and *n* = 2.07 for the second stage.

Karayildirim et al. (2006) [7] used thermogravimetry (TG)/mass spectrometry (MS) to investigate the pyrolysis of PVC at the heating rates of 2, 5, 10, and 15 K/min. Runge–Kutta, and Flynn-Wall-Ozawa (FWO) methods were used to obtain the kinetics parameters. The reported values were *E_a_* = 190 kJ/mol, *A* = 4.40 × 10^7^ min^−1^, and *n* = 1.5 for the first stage, respectively, and *E_a_* = 250 kJ/mol, *A* = 6.57 × 10^7^ min^−1^, and *n* = 1.6 for the second stage.

Wu et al. (2014) [8] studied the pyrolysis and co-pyrolysis of polyethylene (PE), PS, and PVC using TG/Fourier transform infrared (FTIR) at the heating rate of 40 K/min. Pyrolysis of PVC was reported to occur in two stages starting at 594 K and 770 K, respectively.

Yu et al. (2016) [9] reviewed different recycling methods used for the chemical treatment of PVC waste. The onset temperature of PVC pyrolysis was reported to start much earlier than PE, PP, PS, and PET. Two pyrolysis stages of PVC were confirmed. While the first stage was observed in the temperature range of 523–623 K and attributed to the dehydrochlorination of PVC to produce volatile and de-HCl PVC, the second one was reported in the range of 623–798 K and attributed to the pyrolysis of the de-HCl PVC.

Xu et al. (2018) [10] examined the pyrolysis of PVC at high heating rates (100, 300, and 500 K/min) using TGA data. The activation energy for both stages was obtained by three model-free methods namely FWO, Kissinger–Akahira–Sunose (KAS), and Friedman models. For the first stage, the mean *E_a_* values were 48.62, 48.11, and 49.79 kJ/mol obtained by the three methods, respectively, and for the second stage, the obtained values were 113.59, 109.78, and 117.33 kJ/mol, respectively. In addition, two model-fitting methods, namely Coats-Redfern and Criado models, were used to predict the mechanism of the pyrolysis.

Ma et al. (2019) [11] performed a PVC pyrolysis by TG at 20 K/min and two pyrolysis stages were reported. The first stage was observed between 473 and 633 K (dehydrochlorination and polyene chains formation), and the second stage was between 633 and 773 K (degradation of the polyene chains). They used the Coats-Redfern model with the assumption of the first-order reaction mechanism. The values of 114.57 and 7.73 kJ/mol were reported as the activation energy values of both stages, respectively.

Özsin and Pütün (2019) [12] investigated the PVC pyrolysis using between 298 and 1273 K at heating rates of 5, 10, 20, and 40 K/min. To obtain the kinetic parameters, Friedman, FWO, Vyazovkin, and distributed activation energy (DAEM) models were used. Three stages of pyrolysis were reported and attributed to the dehydrochlorination, the formation of alkyl aromatics, and main pyrolysis of PVC, respectively. Activation energy values ranging between 93.2 kJ/mol and 263.7 kJ/mol were reported.

Zhou et al. (2019) [13] studied the pyrolysis of chlorinated polyvinyl chloride (CPVC) using TGA at 10, 20, 30, and 60 K/min heating rates. The FWO model was used to obtain the values of *E_a_* and *A.* However, the Coats-Redfern model was used to predict the reaction mechanism. The reported average values of *E_a_* of Stage I and Stage II were 140.27, and 246.07 kJ/mol, respectively. In addition, F1, and F4 were reported as the most suitable reaction mechanisms of stages I, and II, respectively.

Currently, many researchers are aiming to develop efficient artificial neural network (ANN) models, as an alternative option, to forecast experimental data for different engineering applications. Specifically, ANN modeling has been used to predict the TGA data of the pyrolysis of biomass.

Kinetic data of the thermal decomposition of different materials, including blends, refuse-derived fuel, polycarbonate/CaCO_3_ composites, and high-ash sewage sludge, were predicted by high-efficient ANN models [14,15,16,17,18,19].

Recently, Dubdub and Al-Yaari (2020, and 2021) developed highly efficient ANN models to predict the TGA data of the pyrolysis of LDPE at 5, 10, 20, and 40 K/min [20], pyrolysis of HDPE at 5, 10, 20, and 40 K/min [21], and the co-pyrolysis of PS, PP, LDPE, and HDPE at 60 K/min [22]. In addition, Al-Yaari and Dubdub (2020) [23] developed an ANN model to predict the thermal behavior of the catalytic pyrolysis of HDPE at 5,10, and 15 K/min.

This study aims to build knowledge on PVC pyrolysis using TGA experimental data. The kinetic triplet (activation energy, pre-exponential factor, and reaction mechanism) of the pyrolysis process were obtained by FWO, KAS, Starink, and Coats-Redfern models. In addition, thermodynamic properties of the process of PVC pyrolysis have been investigated. Furthermore, a highly efficient ANN model has been developed to predict the pyrolytic behavior of PVC.

## 2. Materials and Methods

### 2.1. Proximate and Ultimate Analyses

Polymeric materials (PVC) were produced by Ipoh SY Recycle Plastic Sdn. Bhd., Perak, Malaysia. Proximate and ultimate analyses were performed to identify the physicochemical properties of the PVC samples. While the proximate analysis aims to determine the moisture, volatile matter, fixed carbon, and ash contents using Simultaneous Thermal Analyzer STA-6000, manufactured by PerkinElmer, Waltham, MA, USA, the ultimate analysis was performed to determine the % of carbon (C), hydrogen (H), nitrogen (N), and sulfur (S) using 2400 Series II CHNS Elemental Analyzer, manufactured by PerkinElmer, Waltham, MA, USA. Details of both analyses are fully described elsewhere [24].

### 2.2. Thermogravimetry of PVC

PVC pellets were ground into powder by a grinding mill before feeding to the thermogravimetric analyzer. Ten mg of PVC powder samples were used throughout the study. A Thermogravimetric Analyzer TGA-7, manufactured by PerkinElmer Co., Waltham, MA, USA, was used. Thermogravimetric experiments were conducted in an inert atmosphere of pure N_2_ at three different heating rates (5, 10, and 20 K/min).

### 2.3. Determination of the Kinetic Triplet of the PVC Pyrolysis

The reaction kinetics of the PVC pyrolysis can be expressed as follows:(1)$$\frac{d\alpha }{dt}=A\exp\left(-\frac{E_{a}}{RT}\;\right)f\left(\alpha \right)\;\;$$
where α is the reaction conversion, *t* is time, *A* is the pre-exponential (frequency) factor, *E_a_* is the reaction activation energy, *R* is the universal gas constant, *T* is the absolute temperature, and *f(α)* is the conversion-dependent reaction model in its differential form.

For non-isothermal pyrolysis, Equation (1) can be re-written as:(2)$$\beta \frac{d\alpha }{dT}=A\exp\left(-\frac{E_{a}}{RT}\;\right)f\left(\alpha \right)$$
where *β* is the heating rate expressed as the change in temperature with time (dT/dt). If *A*, and *E_a_* are assumed to be independent of α and *f(α)*, *A*, and *E_a_* are independent of *T*. Equation (2) can be integrated and rearranged as follows:(3)$$\int_{0}^{\alpha }\frac{d\alpha }{f\left(\alpha \right)}=\frac{A}{\beta }\int_{T_{o}}^{T}\exp\left(-\frac{E_{a}}{RT}\;\right)\;dT$$
or
(4)$$g\left(\alpha \right)=\frac{A}{\beta }\int_{T_{o}}^{T}\exp\left(-\frac{E_{a}}{RT}\;\right)\;dT\;$$
where $g\left(\alpha \right)$ is the integral form of the conversion-dependent reaction model, and *T_o_* is the initial absolute temperature of the PVC pyrolysis.

The temperature integral does not have an analytical solution, and Equation (4) can be approximated and expressed as:(5)$$g\left(\alpha \right)≅\frac{A\;E_{a}}{\beta \;R}\;P\left(-\frac{E_{a}}{RT}\;\right)$$

Numerical methods need to be used to obtain the polynomial of $\left(-\frac{E_{a}}{RT}\right)$ and a series expansion can give different approximations of the polynomial term.

The FWO model used the following Doyle’s approximation [25] of the temperature integral:(6)$$P\left(-\frac{E_{a}}{RT}\;\right)=Exp\left(-5.331-1.052\;\;\frac{E_{a}}{RT}\right)$$
and thus Equation (5) became:(7)$$g\left(\alpha \right)=\frac{A\;E_{a}}{\beta \;R}\;Exp\left(-5.331-1.052\;\;\frac{E_{a}}{RT}\right)$$
or
(8)$$\ln\beta =\ln\frac{A\;E_{a}}{\;R\;g\left(\alpha \right)}-5.331-1.052\;\;\frac{E_{a}}{RT}$$

However, the KAS model used the Murry-White approximation [26] of the temperature integral where:(9)$$P\left(-\frac{E_{a}}{RT}\;\right)=\frac{Exp\;\left(-\frac{E_{a}}{RT}\right)}{{\left(\frac{E_{a}}{RT}\right)}^{2}}$$
and thus Equation (5) became:(10)$$\ln\frac{\beta }{T^{2}}=\ln\frac{A\;R}{\;E_{a}\;g\left(\alpha \right)}-\frac{E_{a}}{RT}$$

In 2003, Starink [27] used the following approximation of the temperature integral:(11)$$P\left(-\frac{E_{a}}{RT}\;\right)={\left(\frac{T\;R}{E_{a}}\right)}^{1.92}\exp\left(-1.0008\frac{E_{a}}{RT}-0.312\right)$$
and thus Equation (5) became:(12)$$\ln\frac{\beta }{T^{1.92}}=Constant-1.0008\;\frac{E_{a}}{RT}$$
where:(13)$$Constant=\ln\frac{A\;R^{0.92}}{\;E_{a}{}^{0.92}\;g\left(\alpha \right)}-0.312$$

The FWO (Equation (8)), KAS (Equation (10)), and Starink (Equation (12)) models can be expressed in the following general form:(14)$$\ln\frac{\beta }{T^{a}}=b+c\;\frac{E_{a}}{RT}\;$$
where *a*, *b*, and *c* constants are presented in Table 1.

In this work, the values of the apparent activation energy of the PVC pyrolysis have been obtained by plotting $\ln\frac{\beta }{T^{a}}$ vs. $\frac{1}{T}$ for FWO, KAS, and Starink models using the TGA experimental data. The obtained values of *E_a_* by isoconversional methods are independent of the reaction mechanism.

The Coats-Redfern (CR) model, expressed by Equation (15), has been used to determine the most suitable reaction mechanism among 15 solid-state reaction models presented in Table 2. Solid-state kinetic models are categorized based on their mechanistic basis as reaction-order, diffusion, nucleation, and geometrical contraction models [28].

The *E_a_* values obtained by the CR model for different reaction mechanisms were compared with the average *E_a_* values obtained by the isoconversional models. The most appropriate mechanism provides the closest values of activation energy.
(15)$$\ln\frac{g\left(\alpha \right)}{T^{2}}=\ln\frac{A\;R}{\;E_{a}\;\beta }-\frac{E_{a}}{RT}\;\;$$

Then, the values of the pre-exponential factor can be obtained from the slope of the linear relationships of Equations (8), (10), and (12) when the reaction mechanism has been determined. If the reaction mechanism has been determined well, the following linear relationship must be retained (compensation effect).
(16)$$\ln A=dE_{a}+e\;$$
where *d*, and *e* are the compensation parameters that can be obtained from the plot of *lnA* vs. *E_a_*.

### 2.4. Estimation of the Thermodynamic Parameters of the PVC Pyrolysis

Based on the obtained values of activation energy, pre-exponential factor, and the maximum peak temperature (*T_p_*), some of the thermodynamic characteristics of the PVC pyrolysis can be determined using the following Equations:(17)$$\Delta H=E_{a}-R\;T_{p}\;$$
(18)$$\Delta G=E_{a}+R\;T_{p}\ln\left(\frac{k_{B}\;T_{p}}{h\;A}\right)$$
(19)$$\Delta S=\frac{\Delta H-\Delta G}{T_{p}}\;\;\;\;\;\;\;$$
where:
Δ*H*: is the change in enthalpy,Δ*G*: is the change in Gibbs free energy,Δ*S*: is the change in entropy,*T_p_*: is the maximum peak temperature obtained from the derivative thermogravimetric curves,*k_B_*: is the Boltzmann constant (1.381 × 10^−23^ J/K),*h*: is the Planck constant (6.626 × 10^−34^ J/s).

The thermodynamic parameters (ΔH, ΔG and ΔS) are of great importance to the optimization of the large-scale reactor used for pyrolysis.

### 2.5. Performance of Artificial Neural Networks

As mentioned previously, process modeling using artificial neural networks (ANNs) has attracted the attention of researchers due to its robustness, easiness, and cost-effectiveness, especially when the system becomes more complex and non-linear relationships between parameters are adopted.

Typically, the datasets are divided randomly into three subsets: training, validation, and test sets. During the training stage, network learning is established and parameters wight is corrected. However, the network performance is checked during the validation stage and the network is generalized in the test stage [29].

For the best performance of ANNs, a genetic algorithm should be implemented to optimize some topological features such as the number of hidden layers, the number of neurons in the hidden layers, and the transfer functions. The following statistical parameters, expressed by Equations (20)–(23), are used to evaluate the performance of the developed ANN models [30].
(20)$$\mathrm{Correlation}\;\mathrm{coefficient}\;\left(R\right)=\frac{\sum_{i=1}^{N}\left(x_{i}-\bar{x_{i}}\right)\left(y_{i}-\bar{y_{i}}\right)}{\sqrt{\sum_{i=1}^{N}{\left(x_{i}-\bar{x_{i}}\right)}^{2}\sum_{i=1}^{N}{\left(y_{i}-\bar{y_{i}}\right)}^{2}}}$$
(21)$$\mathrm{Root}\;\mathrm{mean}\;\mathrm{square}\;\mathrm{error}\;\left(\mathrm{RMSE}\right)=\sqrt{\frac{1}{N}\;\sum {\left(y-x\right)}^{2}\;}$$
(22)$$\mathrm{Mean}\;\mathrm{absolute}\;\mathrm{error}\;\left(\mathrm{MAE}\right)=\frac{1}{N}\sum \left|y-x\right|$$
(23)$$\mathrm{Mean}\;\mathrm{bias}\;\mathrm{error}\;\left(\mathrm{MBE}\right)=\frac{1}{N}\sum \left(y-x\right)$$
where:
*x*: is the experimental value of the weight left %,*y*: is the predicted value of the weight left %,$\bar{x}$: is the mean values of the experimental weight left %, and$\bar{y}$: is the mean values of the experimental weight left %.

In this investigation, the TGA data of the mass left % during the PVC pyrolysis has been targeted to be predicted by developing an efficient ANN model.

## 3. Results and Discussion

### 3.1. Proximate and Ultimate Analysis

The characterization results of proximate and ultimate analyses of the PVC samples are presented in Table 3. Proximate analysis showed 0.146 wt% of moisture, 88.765 wt% of volatile matter (VM), 10.566 wt% of fixed carbon (FC), and 2.12 wt% of ash. The high valuable contents (VM, and FC) and low ash content indicate the suitability of the production of bioenergy from the pyrolysis of PVC.

In addition, as obtained from the ultimate analysis, low nitrogen and sulfur contents are preferable to avoid the production of toxic gasses such as NO_x_ and SO_x_ and thus benefit the environment.

### 3.2. Thermogravimetry of PVC

The corresponding thermogravimetric (TG) and derivative thermogravimetric (DTG) curves of the PVC pyrolysis at 5, 10, and 20 K/min heating rates are illustrated in Figure 1 and Figure 2, respectively. Although all curves show similarities in their appearance, they were shifted to higher temperatures as the heating rate increased. As the heating rate increased, the mass loss at a specific temperature decreased (see Figure 1). On the other hand, as the heating rate increases, the mass-loss rate increases, and thus the size of the DTG-peak increases (see Figure 2). This finding can be attributed to the thermal lag and/or heat transfer limitations [31].

As shown in both figures, the thermal degradation zone of PVC is in the temperature range of 490 K to 825 K with almost 20 wt% of pyrolysis residues. In addition, both curves revealed two reaction stages covering the temperature ranges of 490–675 K, and 675–825 K, for stages I, and II, respectively. Thus, the PVC pyrolysis occurs a multi-stage mechanism as reported elsewhere [8,9,11,32], The pyrolysis characteristic temperatures of both stages are presented in Table 4. In the first and second degradation stages, characteristic peaks were observed at temperatures of 599 ± 16.4 K. and 724.3 ± 21.9 K, respectively.

While the first reaction stage involves the dehydrochlorination reaction to produce de-HCl PVC and volatiles, the pyrolysis of the de-HCl PVC occurs during the second stage [29]. The first reaction (dehydrochlorination) requires less energy to break the C–Cl bond when compared to the energy required for the second reaction (de-HCl PVC pyrolysis) where the C–C stable bond is broken. Thus, as illustrated in Figure 2, the peak of the first main reaction is much bigger than the peak of the second one which is in full agreement with the available literature [10].

The apparent activation energy has been obtained using TGA data along with FWO, KAS, and Starink models, expressed by Equations (8), (10), and (12), respectively. From the slope of the plots of $\left(\ln\frac{\beta }{T^{a}}\right)$ versus ($\frac{1}{T})$, where the exponent *a* is defined in Table 1, the values of *E_a_* were determined at a conversion range of 0.1–0.8. While stage I covers the conversion range of 0.1–0.6, stage II covers the range of 0.7–0.8 and this is in full agreement with published data [10]. Regression lines of all plots for both pyrolytic stages are presented in Figure 3 and Figure 4, respectively, and the values of *E_a_* obtained by all models are presented in Table 5.

As shown in Figure 3 and Figure 4, although all regression lines for the FWO, KAS, and Starink models are parallel, the gap between those of stage I is smaller than those of stage II which indicates that the reaction that occurs in stage I (dehydrochlorination) is faster than that of stage II (de-HCl PVC pyrolysis). In addition, the gap between the regression lines of stage I increased as the conversion increased, which indicates a decrease in the conversion rate of the dehydrochlorination process with time. This is also confirmed by the TG and DTG curves (Figure 1 and Figure 2, respectively) when the plateau (shoulder) zone between the first and the second stages is approached. Furthermore, as the gap between the regression lines increased, a higher variation in the value of activation energy was observed (see Table 5).

As shown in Table 5, TGA data were fitted well (R^2^ > 0.95) by the FWO, KAS, and Starink models and the obtained values were very close and thus indicate that all three models are suitable to be used. The average value of activation energy of the first stage of the PVC pyrolysis is 75 kJ/mole with a regression coefficient (R^2^) of 0.9702 and that of the second stage is 140 kJ/mole with an R^2^ value of 0.9902.

For stage I, at low conversion (α < 0.3), the obtained values of *E_a_* are larger than the average value of *E_a_* which can be attributed to the low energy provided initially to the dehydrochlorination reaction. However, at high conversion (α > 0.5), the provided energy is higher than the required one and thus there is a reduction in the *E_a_* values. Similarly, for stage II, the *E_a_* value at α = 0.7 is less than that at α = 0.8 which could be explained using the same concept. However, as mentioned earlier, the *E_a_* values of stage II are larger than those of stage I which are due to the difference in the amount of the required energy for reactions that occur in both stages. A similar trend was reported earlier for the PVC pyrolysis by Mumbach et al. [33], but with different values which can be attributed to the difference in the PVC compositions of both works.

Then, the Coat-Redfern model, expressed by Equation (15), was used to obtain the most suitable reaction mechanism/s for both stages of the PVC pyrolysis. Values of activation energy and pre-exponential factor at different heating rates for 15 solid-state reaction mechanisms, defined in Table 2, were obtained from the slope and the intercept of the plots of $\ln\frac{g\left(\alpha \right)}{T^{2}}$ versus $\frac{1}{T}$. The obtained kinetic parameters are presented in Table 6 and Table 7 for stage I, and stage II, respectively.

As shown in Table 6 and Table 7, the Coats-Redfern model fitted well the TGA data of the PVC pyrolysis with a regression coefficient of almost R^2^ > 0.99. The average *E_a_* values obtained by Coats-Redfern were then compared with the average values of *E_a_* obtained by the isoconversional models for both stages. The most suitable reaction mechanism should give a closer value of the activation energy. Based on this criterion, stage I was suitably represented by the power-law nucleation reaction model (P2) (*E_a_* = 79 kJ/mol, R^2^ = 0.9962), and stage II was best represented by the 3rd order reaction mechanism (F3) (*E_a_* = 125 kJ/mol, R^2^ = 0.9986).

The power-law (P2) reaction model is among the simplest cases of nucleation models where the growth of the reaction nuclei is assumed constant, and the reaction rate follows the power law ($f\left(\alpha \right)=2\alpha^{1/2})$ [28]. However, in reaction-order models, the reaction rate is directly proportional to the reactants remaining fraction raised to the reaction order. For F3, $f\left(\alpha \right)={\left(1-\alpha \right)}^{3}$.

After the determination of the most suitable reaction mechanisms for both stages, the values of the pre-exponential factor were calculated by the isoconversional models. These values are presented in Table 8. The obtained values by isoconversional models are comparable with those obtained by Coats-Redfern for the selected reaction mechanism. Alternatively, compensation effect parameters (*d* = 0.1244, and *e* = 4.9338) can be obtained from the regression line of Figure 5. Then, the pre-exponential factor can be calculated using Equation (16) for each value of activation energy obtained by the isoconversional models.

To check the proposed mechanism, the Criado model [24] or compensation effect (i.e., Equation (16)) [34,35] can be used. In this work, the linearity between *lnA* and *E_a_* was checked at both stages. As shown in Figure 5, a linear relationship was confirmed with a regression coefficient of 0.9865. This implies the suitability of the proposed reaction models for both stages, and the average values of *A* were 1.79 × 10^6^ min^−1^, and 4.84 × 10^9^ min^−1^ for stages I, and II, respectively.

### 3.3. Estimation of the Thermodynamic Parameters of the PVC Pyrolysis

Thermodynamic properties along with the kinetic triplet are very important to the optimization of the large-scale pyrolytic reactor. Therefore, the thermodynamic parameters (ΔH, ΔG and ΔS) were obtained at different heating rates (5, 10, and 20 K/min) for both stages of the PVC pyrolysis as presented in Table 9.

As presented in Table 9, the PVC pyrolysis has positive ΔH values (stage I: 70.4 ± 0.17 kJ/mol, and stage II: 134 ± 0.22 kJ/mol). The positive sign of ΔH values indicates that both stages include endothermic reactions. In addition, these results reveal that higher energy is needed for stage II when compared with stage I. Furthermore, a small energy barrier (*E_a_*_–_ΔH) of (stage I: 4.6 kJ/mole, and stage II: 6 kJ/mole) was observed. This amount of energy must be added for the reactions to take place.

Moreover, the positive and negative signs of ΔG and ΔS, respectively, indicate that the PVC pyrolysis is a nonspontaneous process (i.e., products have a lower disorder degree than the reactants). Additionally, the values of ΔS can indicate the reactivity order (stage I presented a slightly lower reactivity when compared to stage II). Furthermore, the positive values of ΔG reflect the amount of available bioenergy that can be produced from the PVC pyrolysis in each stage. The thermodynamic results reveal the promising potential of the PVC pyrolysis to efficiently produce bioenergy.

### 3.4. Pyrolysis Prediction by ANN Model

The experimental TGA datasets (403 datasets) were automatically and randomly divided into three sets: 70% (283 data sets) were used for training, 15% (60 data sets) were used for validation, and 15% (60 data sets) were used for testing.

To find the best topology of the ANN model aiming to predict the TGA data of the PVC pyrolysis, the number of hidden layers, number of neurons in each layer, and transfer functions have been optimized. Table 10 shows the performance of different ANN structures. The value of correlation coefficient (R) was considered as the main criterion for the selection of the most efficient network structure to estimate the weight left % as the output variable. As presented in Table 10, the ANN7 model shows the best performance (R = 0.99999) with the minimum no. of hidden layers and neurons, and the topology of the selected network (2-10-10-1) is presented in Figure 6. The ANN7 model has two input parameters (temperature, and heating rate), two hidden layers having 10 neurons in each layer, and an output parameter (PVC weight left %). In addition, the model has a feed-forward back-propagation characteristic and the TANSIG-LOGSIG transfer function was recommended.

Then, the performance of the developed model was tested. As shown in Figure 7, a full agreement between the ANN-predicted values (*Y*-axis) and the experimental values (*X*-axis) has been guaranteed (R = 1.0). In addition, as presented in Table 11, RMSE, MAE, and MBE were significantly low. This implies the robustness of the developed model to predict the TGA data of the PVC pyrolysis.

Additionally, the performance of the developed ANN model was checked using new datasets. This step is known as ‘the simulation step’. For this purpose, nine extra datasets were utilized. Table 12 presents the input and the targeted-output data of this step.

As presented in Figure 8, the full agreement between the experimental and the predicted values indicates the high performance of the developed model. In addition, the R-value of one and very low values of RMSE, MAE, and MBE (see Table 13) were obtained and additionally confirmed the robustness of the developed model to predict the TGA data of the PVC pyrolysis.

## 4. Conclusions

In this work, a comprehensive investigation of the pyrolysis of PVC at heating rates of 5, 10, and 20 K/min using thermogravimetric analysis and artificial neural network modeling was performed. Three isoconversional integral models (FWO, KAS, and Starink) and the Coats-Redfern non-isoconversional model were used to obtain the kinetic triplet (activation energy, pre-exponential factor, and reaction mechanism) of the PVC pyrolysis. Based on the reported results, the following conclusions can be drawn:The proximate analysis of PVC samples indicates the suitability of the production of bioenergy from the pyrolysis of PVC. For future work, products of the PVC pyrolysis should be identified, and their caloric values are to be obtained and compared with those of conventional fuels.The ultimate analysis of PVC samples showed low nitrogen and sulfur contents which are preferable to avoid the production of toxic gasses such as NOx and Sox, and thus benefit the environment.Thermogravimetric and derivative-thermogravimetric curves revealed that the PVC pyrolysis occurred in two stages covering the temperature range of 490–825 K.The kinetic triplets of both stages of the PVC pyrolysis were obtained and thus step-by-step guidance was outlined. This procedure can be followed to obtain the kinetic triplet of the pyrolysis of different wastes.The thermodynamic properties (ΔH, ΔG, and ΔS) of the process of the PVC pyrolysis showed that the reactions of both stages are endothermic and nonspontaneous, and confirmed the suitability of the production of bioenergy by the pyrolysis process.A highly efficient ANN model to predict the TGA data of the PVC pyrolysis was developed. It has the following characteristics: a feed-forward back-propagation algorithm, TANSIG-LOGSIG transfer function, and 2-10-10-1 network topology. For future studies, other artificial intelligence (AI) algorithms can be developed to predict the TGA data of the PVC pyrolysis, and their performance can be tested.

## Figures and Tables

**Figure 1 polymers-13-04359-f001:**
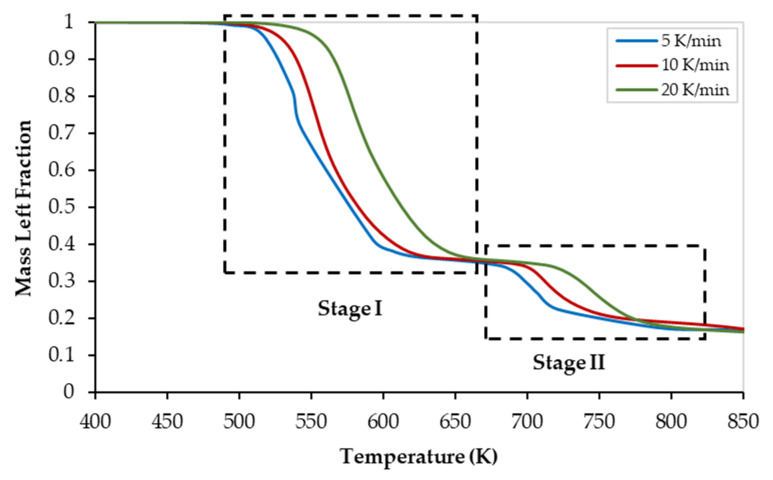
Thermogravimetric (TG) curves of the PVC pyrolysis at different heating rates.

**Figure 2 polymers-13-04359-f002:**
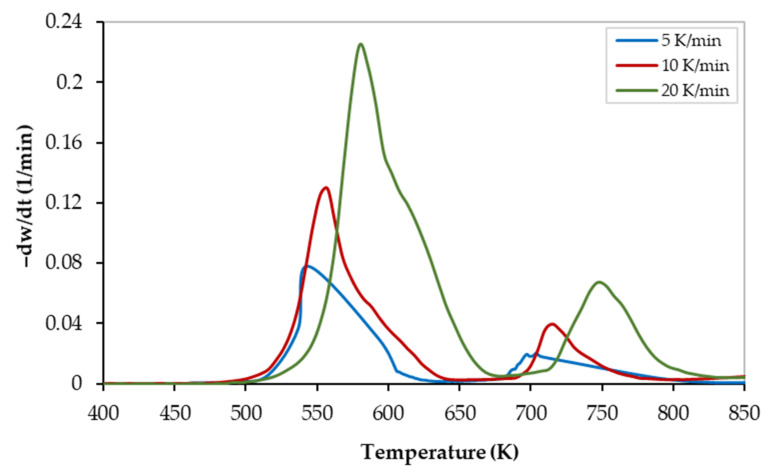
Derivative-thermogravimetric (DTG) curves of the PVC pyrolysis at different heating rates.

**Figure 3 polymers-13-04359-f003:**
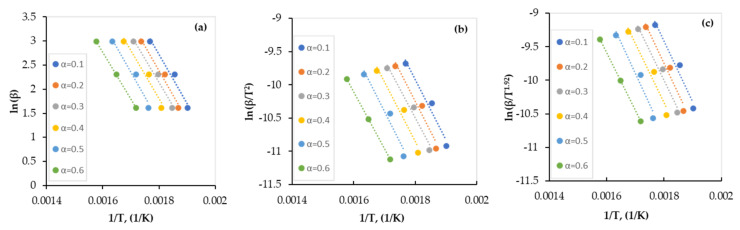
Regression lines of the experimental data of the PVC pyrolysis (stage I) by: (**a**) FWO, (**b**) KAS, and (**c**) Starink models.

**Figure 4 polymers-13-04359-f004:**
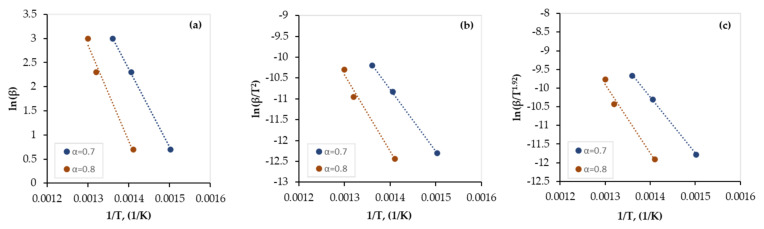
Regression lines of the experimental data of the PVC pyrolysis (stage II) by: (**a**) FWO, (**b**) KAS, and (**c**) Starink models.

**Figure 5 polymers-13-04359-f005:**
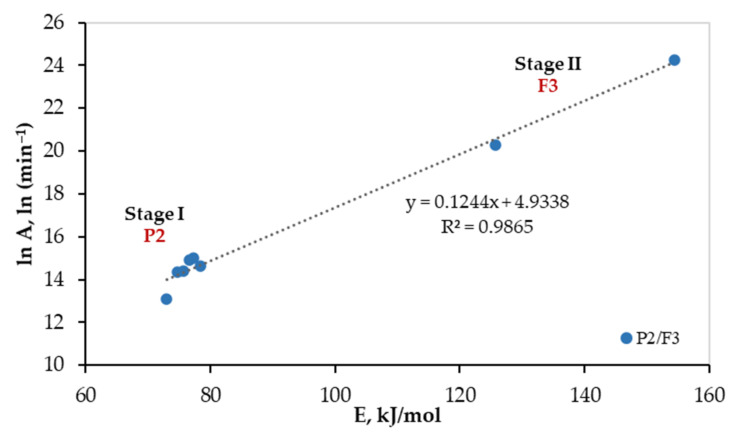
Linear fitted curve for the compensation effect.

**Figure 6 polymers-13-04359-f006:**
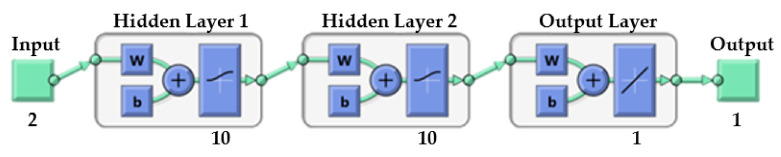
Topology of the best-selected network.

**Figure 7 polymers-13-04359-f007:**
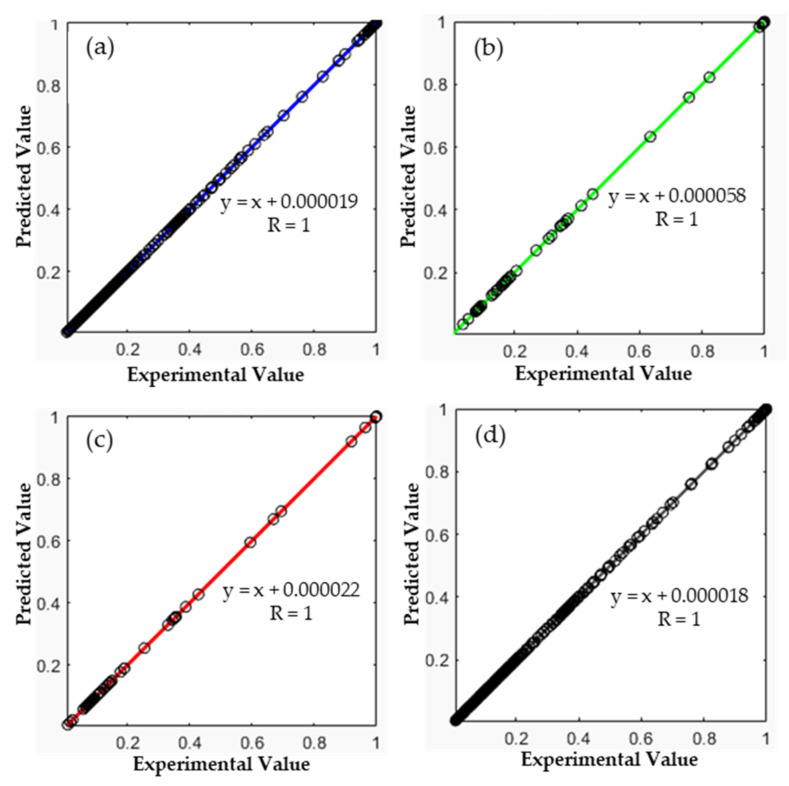
Regression plots of (**a**) training data, (**b**) validation data, (**c**) test data, and (**d**) complete datasets of the (2-10-10-1) ANN model.

**Figure 8 polymers-13-04359-f008:**
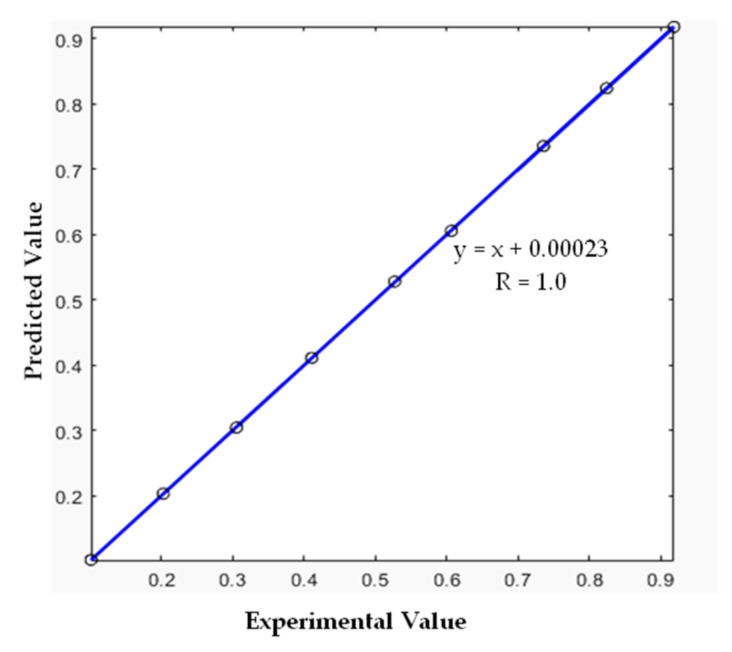
Comparison between predicted and experimental data using the simulation data.

**Table 1 polymers-13-04359-t001:** Parameters of the generalized form of the isoconversional models.

Model	*a*	*b*	*c*
FWO	0	$ln\frac{A\;E_{a}}{\;R\;g\left(\alpha \right)}-5.331$	−1.052
KAS	2	$ln\frac{A\;R}{\;E_{a}\;g\left(\alpha \right)}$	−1
Starink	1.92	$ln\frac{A\;R^{0.92}}{\;E_{a}{}^{0.92}\;g\left(\alpha \right)}-0.312$	−1.0008

**Table 2 polymers-13-04359-t002:** List of the most used solid-state reaction models.

Model	Mechanism	$g\left(\alpha \right)$
**Reaction-Order Models**		
F1	First-order reaction	$-\ln\left(1-\alpha \right)$
F2	Second-order reaction	${\left(1-\alpha \right)}^{-1}-1$
F3	Third-order reaction	$[{\left(1-\alpha \right)}^{-1}-1]/2$
**Diffusion Models**		
D1	One-dimensional diffusion	$\alpha^{2}$
D2	Two-dimensional diffusion	$\left(1-\alpha \right)\ln\left(1-\alpha \right)+\alpha$
D3	Three-dimensional diffusion	${\left[1-{\left(1-\alpha \right)}^{1/3}\right]}^{2}$
**Nucleation Models**		
P2	Power law ($n=\frac{1}{2}$)	$\alpha^{1/2}$
P3	Power law ($n=\frac{1}{3}$)	$\alpha^{1/3}$
P4	Power law ($n=\frac{1}{4}$)	$\alpha^{1/4}$
A2	Avrami-Erofeev ($n=\frac{1}{2}$)	${\left[-\ln\left(1-\alpha \right)\right]}^{1/2}$
A3	Avrami-Erofeev ($n=\frac{1}{3}$)	${\left[-\ln\left(1-\alpha \right)\right]}^{1/3}$
A4	Avrami-Erofeev ($n=\frac{1}{4}$)	${\left[-\ln\left(1-\alpha \right)\right]}^{1/4}$
**Geometrical Contraction Models**		
R1	Prout-Tompkins	α
R2	Contracting cylinder	1−${\left(1-\alpha \right)}^{1/2}$
R3	Contracting sphere	1−${\left(1-\alpha \right)}^{1/3}$

**Table 3 polymers-13-04359-t003:** Some characteristics of the PVC samples.

Proximate Analysis, wt%				Ultimate Analysis, wt%				
Moisture	Volatile Matter	Fixed Carbon	Ash	C	H	N	S	O ^a^
0.146	88.765	10.566	0.523	83.75	13.70	0.14	0.78	1.63

^a^ by difference.

**Table 4 polymers-13-04359-t004:** Characteristic temperatures of the PVC pyrolysis at different heating rates.

Heating Rate (K/min)	Stage I				Stage II			
	On-Set (K)	Peak (K)	Final (K)	Mass Loss (%)	On-Set (K)	Peak (K)	Final (K)	Mass Loss (%)
5	490	540	630	64	675	700	810	82
10	495	557	640	64	690	720	815	82
20	500	580	675	64	720	753	825	82
Average *	495 ± 4.08	559 ± 16.4	648.3 ± 19.3	64	695 ± 18.7	724.3 ± 21.9	816.7 ± 6.2	82

* mean ± standard deviation Determination of the kinetic triplet of the PVC pyrolysis.

**Table 5 polymers-13-04359-t005:** Activation energy values obtained by isoconversional models.

Conversion	FWO		KAS		Starink		Average	
	*E_a_* (kJ/mol)	*R* ^2^	*E_a_* (kJ/mol)	*R* ^2^	*E_a_* (kJ/mol)	*R* ^2^	*E_a_* (kJ/mol)	*R* ^2^
**Stage I**								
0.1	80	0.9704	75	0.9628	75	0.9631	77	0.9654
0.2	81	0.9708	75	0.9631	76	0.9634	76	0.9658
0.3	78	0.9752	73	0.9681	73	0.9685	73	0.9706
0.4	79	0.9635	74	0.9533	74	0.9538	74	0.9569
0.5	82	0.9682	76	0.9595	77	0.9599	77	0.9625
0.6	77	1	71	1	71	1	71	1.0000
Average	79	0.9747	74	0.9678	74	0.9681	75	0.9702
**Stage II**								
0.7	129	0.9995	124	0.9993	124	0.9993	125	0.9994
0.8	157	0.9827	153	0.98	153	0.9802	154	0.9810
Average	143	0.9911	138	0.9897	139	0.9898	140	0.9902

**Table 6 polymers-13-04359-t006:** Kinetic parameters obtained by Coats-Redfern model for Stage I.

Reaction Mechanism	Heating Rates									Average		
	5 K/min			10 K/min			20 K/min					
	E_a_	*ln* *A*	*R* ^2^	*E_a_*	*ln* *A*	*R* ^2^	*E_a_*	*ln* *A*	*R* ^2^	*E_a_*	*ln* *A*	*R* ^2^
	(kJ/mol)	*ln* (min^−1^)		(kJ/mol)	*ln* (min^−1^)		(kJ/mol)	*ln* (min^−1^)		(kJ/mol)	*ln* (min^−1^)	
F1	187	38.66	0.9999	173	36.03	0.9971	182	36.73	0.9981	181	37	0.9984
F2	202	42.25	0.9999	189	39.63	0.9988	198	40.26	0.9991	196	41	0.9993
F3	218	46.06	0.9991	205	43.44	0.9996	215	43.98	0.9997	213	44	0.9995
D1	355	75.25	0.9992	327	68.49	0.9947	343	69.31	0.9967	342	71	0.9969
D2	364	76.69	0.9995	336	69.94	0.9957	353	70.74	0.9973	351	72	0.9975
D3	374	77.4	0.9998	346	70.67	0.9965	363	71.45	0.9977	361	73	0.9980
D4	368	75.93	0.9996	339	69.18	0.996	356	69.98	0.9974	354	72	0.9977
A2	89	16.67	0.9999	82	16.09	0.9968	86	16.76	0.9978	86	17	0.9982
A3	57	9.9	0.9999	52	12.85	0.9968	54	13.66	0.9975	54	12	0.9981
A4	89	16.67	0.9999	37	15.71	0.9958	38	16.51	0.9972	55	16	0.9976
R1	173	35.31	0.9991	159	32.67	0.9944	167	33.4	0.9965	166	34	0.9967
R2	180	36.27	0.9996	166	33.63	0.9959	174	34.35	0.9974	173	35	0.9976
R3	183	36.43	0.9997	168	33.79	0.9963	177	34.5	0.9976	176	35	0.9979
P2	82	14.96	0.999	75	14.37	0.9936	79	15.05	0.9961	79	15	0.9962
P3	52	10.9	0.9989	47	13.84	0.9927	49	14.64	0.9955	49	13	0.9957
P4	37	13.83	0.9987	33	16.41	0.9916	35	17.24	0.9948	35	16	0.9950

**Table 7 polymers-13-04359-t007:** Kinetic parameters obtained by Coats-Redfern model for Stage II.

Reaction Mechanism	Heating Rates									Average		
	5 K/min			10 K/min			20 K/min					
	*E_a_*	*ln* *A*	*R* ^2^	*E_a_*	*ln* *A*	*R* ^2^	*E_a_*	*ln* *A*	*R* ^2^	*E_a_*	*ln* *A*	*R* ^2^
	(kJ/mol)	*ln* (min^−1^)		(kJ/mol)	*ln* (min^−1^)		(kJ/mol)	*ln* (min^−1^)		(kJ/mol)	*ln* (min^−1^)	
F1	41	14.81	0.9981	31	18.23	0.9981	29	19.34	0.9987	34	17	0.9983
F2	93	14.35	0.9978	63	12.81	0.999	63	13.73	0.9992	73	14	0.9987
F3	162	45.76	0.9977	105	18.02	0.999	108	19.01	0.999	125	28	0.9986
D1	31	17.17	0.9989	31	19.11	0.9959	50	20.63	0.9977	37	19	0.9975
D2	45	15.33	0.9989	42	17.93	0.9972	59	19.52	0.998	49	18	0.9980
D3	67	12.91	0.9987	57	16.8	0.9982	69	18.4	0.9975	64	16	0.9981
D4	52	15.57	0.9988	47	18.58	0.9977	62	20.17	0.997	54	18	0.9978
A2	15	18.55	0.9198	10	20.81	0.9134	8	21.7	0.9959	11	20	0.9430
A3	6	19.21	0.9901	2	20.45	0.9635	1	21.37	0.9427	3	20	0.9654
A4	1	18.21	0.9964	1	20.38	0.9948	2	21.92	0.9788	1	20	0.9900
R1	10	19.46	0.9973	10	21.26	0.9888	7	22.12	0.9976	9	21	0.9946
R2	22	18.38	0.9981	19	20.76	0.9962	16	21.87	0.9965	19	20	0.9969
R3	28	17.97	0.9981	23	20.63	0.9971	20	21.76	0.9825	24	20	0.9926
P2	1	18.86	0.9455	1	20.6	0.8186	3	15.05	0.9656	2	18	0.9099
P3	5	21.04	0.9988	5	22.75	0.9939	6	14.64	0.9964	5	19	0.9964
P4	1	18.86	0.9996	7	23.36	0.998	7	17.24	0.9986	5	20	0.9987

**Table 8 polymers-13-04359-t008:** Pre-exponential factor values obtained by isoconversional models.

Conversion	*ln* [*A* (min^−1^)]			
	FWO	KAS	Starink	Average
**Stage I** (P2 reaction mechanism)				
0.1	16.0	14.3	14.5	14.9
0.2	16.1	14.4	14.6	15.0
0.3	15.5	13.7	13.9	14.3
0.4	15.5	13.7	13.9	14.4
0.5	15.8	14.0	14.2	14.7
0.6	14.3	12.4	12.6	13.1
Average	15.5	13.8	14.0	14.4
**Stage II** (F3 reaction mechanism)				
0.7	21.0	19.8	20.0	20.3
0.8	24.8	23.9	24.1	24.3
Average	22.9	21.85	22.05	22.3

**Table 9 polymers-13-04359-t009:** Thermodynamic parameters estimated for the process of the pyrolysis of PVC.

Stage	I			II		
Heating rates (K/min)	5	10	20	5	10	20
**Kinetic Parameters**						
*E_a_* (kJ/mol)	75			140		
*A* (min^−1^)	1.79 × 10^6^			4.84 × 10^9^		
Kinetic Equation	$\frac{d\alpha }{dt}=1.79\times 10^{6}\;e^{\frac{-75000}{R\;T}}\;\left(2\alpha^{\frac{1}{2}}\right)$			$\frac{d\alpha }{dt}=4.84\times 10^{9}\;e^{\frac{-140000}{R\;T}}\;{\left(1-\alpha \right)}^{3}$		
*T_p_* (K)	540	557	580	700	720	753
**Thermodynamic Parameters**						
ΔH (kJ/mol)	70.5	70.4	70.2	134.2	134.0	133.7
ΔG (kJ/mol)	145.2	147.6	150.8	186.6	188.1	190.6
ΔS (kJ/mol.K)	−0.14	−0.14	−0.14	−0.07	−0.08	−0.08
**Potential Energy Barrier**						
*E_a_*_–_ΔH (kJ/mol) *	4.6			6		

* Based on the mean values of ΔH

**Table 10 polymers-13-04359-t010:** Prediction performance of different ANN structures.

Model	Network Topology	1st Transfer Function (Hidden Layer 1)	2nd Transfer Function (Hidden Layer 2)	R
ANN1	NN-2-5-1	TANSIG	-	0.99885
ANN2	NN-2-5-1	LOGSIG	-	0.99808
ANN3	NN-2-10-1	TANSIG	-	0.99956
ANN4	NN-2-10-1	LOGSIG	-	0.99940
ANN5	NN-2-15-1	TANSIG	-	0.99957
ANN6	NN-2-15-1	LOGSIG	-	0.99985
ANN7	NN-2-10-10-1	TANSIG	LOGSIG	0.99999
ANN8	NN-2-10-10-1	LOGSIG	LOGSIG	0.99996
ANN9	NN-2-10-15-1	LOGSIG	LOGSIG	0.99999
ANN10	NN-2-15-10-1	LOGSIG	LOGSIG	0.99979
ANN11	NN-2-10-15-1	TANSIG	LOGSIG	0.99996
ANN12	NN-2-15-15-1	LOGSIG	TANSIG	0.99988
ANN13	NN-2-15-15-1	LOGSIG	LOGSIG	0.99999

**Table 11 polymers-13-04359-t011:** Statistical parameters of the (2-10-10-1) ANN network.

Set	Statistical Parameters			
	R	RMSE	MAE	MBE
Training	1.00000	0.000507	0.000315	0.000037
Validation	1.00000	0.000547	0.000319	0.000026
Test	1.00000	0.000388	0.000255	0.000005
All	1.00000	0.000481	0.000296	0.000027

**Table 12 polymers-13-04359-t012:** Input and targeted-output data used during the simulation step.

No.	Input Data		Targeted-Output Data
	Heating Rate (K/min)	Temperature (K)	Weight Left (Fraction)
1	2	540.5	0.74
2	2	599.2	0.41
3	2	758.6	0.10
4	10	547.1	0.82
5	10	577.8	0.53
6	10	761.3	0.20
7	20	563.0	0.92
8	20	596.5	0.61
9	20	735.5	0.31

**Table 13 polymers-13-04359-t013:** Statistical parameters of the (2-10-10-1) model during the simulation step.

Set	Statistical Parameters			
	R	RMSE	MAE	MBE
Simulated	1.00000	0.000576	0.000479	−0.000118

## Data Availability

The authors confirm that the data supporting the findings of this study are available within the article.

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
