# Peer review of "Pyrolytic Behavior of Polyvinyl Chloride: Kinetics, Mechanisms, Thermodynamics, and Artificial Neural Network Application"

_polymers, 2021, doi:10.3390/polym13244359_

Round 1

Reviewer 1 Report

The issues of polyvinyl chloride processing are still topical. This material raises a lot of controversy due to its properties and the effects of use and the risk it carries during decomposition during combustion. Methods based on the pyrolysis of waste polyvinyl chloride presence are considered to be highly efficient treatment methods. The present work is very interesting and gives a different perspective on the problem of pyrolysis. The use of artificial neural networks makes it possible to predict the results of the process.

  1. The introduction is extensive and fully covers the presented problem.
  2. The literature review is relevant and up-to-date.
  3. The results presented in the form of tables and figures are well described.
  4. The adopted model of artificial neural networks is correct. In the future, it would be worth testing other network models and checking how they behave and what the answer is.

Author Response

Dear Respected Reviewer,

Greetings

Thanks for your time and kind effort paid for the review process of this manuscript.

Best Regards

Reviewer 2 Report

The main recommendations for improving the article are the following:

Section 3 Results and Discussion needs improvements because it contains many figures and tables, which are not discussed enough.
The results must be interpretive rather than descriptive and connect the research results with relevant literature citations for validity and reliability.
The discussions are not well-presented as it does not integrate with the research study results to provide a coherent argument.
The Conclusion section should include the key focus of the study in the conclusion. The conclusions are not supported by the research data, indicating a more straightforward path for future studies on the topic. A follow-up of restating results with supporting literature reviews could make the conclusion section more effective.

Good luck!

Author Response

Dear Respected Reviewer,

Greetings

Many thanks for your kind valuable comments that lead to a significant improvement of the manuscript.

Please find attached the authors response to your comments. In addition, please check the revised manuscript.

Best Regards

Round 2

Reviewer 2 Report

Good luck!